# Effects of Short-Term Structural Exercise on Cardiopulmonary Function, Quality of Life, and Oxidative Status in Liver Transplant Recipients: A Case Series

**DOI:** 10.3390/jfmk10030313

**Published:** 2025-08-14

**Authors:** Narubet Mekkhayai, Jirakrit Leelarungrayub, Supatcha Konghakote, Rungtiwa Kanthain, Khanittha Wonglangka, Sunhawit Junrungsee, Mujalin Prasannarong

**Affiliations:** 1Sports Science, Multidisciplinary and Interdisciplinary School (MIdS), Chiang Mai University, Chiang Mai 50200, Thailand; narubet.me@gmail.com (N.M.); donrawee.leela@cmu.ac.th (J.L.); 2Department of Physical Therapy, Faculty of Associated Medical Sciences, Chiang Mai University, Chiang Mai 50200, Thailand; supatcha.k@cmu.ac.th (S.K.); rungtiwa.k@cmu.ac.th (R.K.); khanittha.w@cmu.ac.th (K.W.); 3Department of Surgery, Faculty of Medicine, Chiang Mai University, Chiang Mai 50200, Thailand; sunhawit.j@cmu.ac.th

**Keywords:** rehabilitation, hospitalization, inspiratory muscle training, early mobilization, oxidative stress, health

## Abstract

**Background:** Living donor liver transplantation (LDLT) poses significant physiological challenges, especially during early postoperative recovery. While the long-term benefits of structured rehabilitation are well documented, data on short-term effects—particularly during the critical early inpatient phase—remain limited. This study aimed to evaluate the short-term impact of a structured exercise program on cardiopulmonary function, respiratory muscle strength, physical performance, oxidative stress markers, and quality of life in LDLT recipients. **Methods:** Four LDLT recipients (2 males, 2 females; mean age 48.00 ± 18.35 years) underwent a 4-week inpatient rehabilitation protocol. Weeks 1–2 involved conventional care, while weeks 3–4 included structured exercise consisting of early mobilization and inspiratory muscle training. Outcome measures included cardiopulmonary exercise testing (CPET), spirometry, maximal inspiratory and expiratory pressures (PImax, PEmax), 6 min walk distance (6MWD), lower limb muscle strength, Chronic Liver Disease Questionnaire (CLDQ), and serum oxidative stress markers (total antioxidant capacity [TAC] and malondialdehyde [MDA]). **Results:** All patients demonstrated postoperative declines in VO_2_ peak, PImax, PEmax, and TAC. Structured exercise yielded clinically meaningful improvements in respiratory muscle strength, notably in female and younger participants. Two younger patients showed increased 6MWD; however, no patient regained preoperative VO_2_ peak. TAC levels decreased following the intervention, and MDA levels remained stable in most cases. **Conclusions:** A two-week structured exercise program during early postoperative recovery may provide partial benefits in respiratory muscle strength and physical performance but is insufficient to restore full cardiopulmonary function in LDLT recipients. Longer rehabilitation periods may be necessary to achieve preoperative recovery levels.

## 1. Introduction

Liver transplantation (LT) is a life-saving treatment for patients with end-stage liver disease and acute liver failure. When supportive or medical therapies are no longer sufficient, transplantation becomes the only definitive intervention that can restore hepatic function and improve long-term survival [1]. Liver transplantation has been associated with significant physiological stress, particularly in the cardiopulmonary system. Patients with cirrhosis often exhibit cirrhotic cardiomyopathy, elevated pulmonary pressure, and impaired oxygen utilization, which may worsen perioperatively [2,3]. In the early postoperative phase, fatigue, respiratory muscle weakness, and lower-limb muscle atrophy are common and may lead to physical deconditioning [4,5]. Reperfusion injury after LT is associated with lowered antioxidant defenses and evidence of free radical damage. Total antioxidant capacity (TAC) in LT recipients significantly decreased after LT [6]. Malondialdehyde (MDA), an indicator of oxidative stress, increases compared to healthy controls [7]. Even months after surgery, recipients often demonstrate suboptimal cardiopulmonary function, including reduced peak oxygen uptake (VO_2_ peak), exercise intolerance, and muscle weakness [8,9]. These limitations not only delay mobilization but may also compromise long-term outcomes.

Exercise-based rehabilitation has been increasingly recognized as essential for recovery, encompassing the preoperative, early postoperative, and late rehabilitation phases [10,11]. The first 7 days after LT represent a critical window in which physiological decline is most pronounced, as evidenced by the high incidence of early allograft dysfunction and hemodynamic instability [12,13]. Early mobilization programs, initiated during the early postoperative period, have gained increasing recognition for their role in enhancing recovery. A randomized controlled trial demonstrated that initiating rehabilitation within the intensive care unit led to earlier achievement of functional milestones such as sitting and walking, improved muscle strength and exercise capacity, and reduced fatigue, all without adverse effects [14]. However, most rehabilitation programs are not initiated during the critical window of physiological decline. Standard care frequently delays bedside mobilization until approximately 7–10 days after surgery [15,16]. Exercise interventions for 8 to 24 weeks are implemented after ICU discharge and clinical stability is achieved, including significant improvements in aerobic capacity (e.g., VO_2_ peak), muscle strength, functional mobility, and fatigue [15,17,18]. In addition, inspiratory muscle training (IMT) has been shown to reduce postoperative pulmonary complication rates and significantly improve respiratory muscle function in patients undergoing upper abdominal surgery [19]. This may suggest potential benefits for postoperative LT recipients. Together, exercise-based rehabilitation and IMT have individually demonstrated significant benefits in improving cardiopulmonary function in LT recipients. To date, no study has systematically examined the combination of these two modalities in this population, particularly during the early postoperative recovery phase. This represents an important gap in the current literature that warrants further investigation.

Therefore, this study aimed to investigate the short-term effects of a structured exercise program comprising early rehabilitation and IMT on cardiopulmonary function in hospitalized LT recipients. We also examined trends in cardiopulmonary recovery before and after LT to assess the potential benefits of treatment initiated during hospitalization.

## 2. Materials and Methods

### 2.1. Study Population and Data Collection

Four patients underwent living donor LT (LDLT) at Maharaj Nakorn Chiang Mai Hospital between 1 February 2024, and 31 January 2025. Eligible participants were aged ≥20 years, diagnosed with end-stage liver disease (ESLD), and deemed medically stable for early postoperative rehabilitation by their attending physicians. All the participants provided written informed consent. Exclusion criteria included medical contraindications to physical testing (e.g., inability to perform cardiopulmonary exercise testing, chronic respiratory diseases, or neurological conditions that compromised exercise safety or performance). Baseline demographic data were obtained from hospital charts.

This study was conducted in accordance with the Declaration of Helsinki (2013), Good Clinical Practice (ICH-GCP E6[R2]), and the ethical guidelines for human research issued by the Faculty of Medicine, Chiang Mai University. The study protocol was reviewed and approved by the Research Ethics Committee of the Faculty of Medicine of Chiang Mai University (Approval Code: NONE-2566-0422).

### 2.2. Cardiopulmonary Exercise Testing (CPET)

CPET was performed using a cycle ergometer with continuous 12-lead ECG and breath-by-breath gas analysis (UltimaTM CPX/CardiO_2_ system, Lode B.V., Groningen, The Netherlands). After a no-load warm-up phase, a ramp protocol of 4 W/kg/min was implemented using a cycle ergometer [20]. Patients pedaled until reaching voluntary exhaustion, limiting symptoms, arrhythmias, or 80% of their age-predicted maximum heart rate (HRmax). This submaximal threshold was chosen to ensure safety while enabling accurate VO_2_ peak assessment (mL/kg/min). In liver transplant candidates with poor physical status or comorbidities, 80% predicted HRmax is recommended as a safe cutoff [21], and for deconditioned participants, termination points may be further modified for safety.

### 2.3. Six-Minute Walk Distance (6MWD)

Functional capacity was assessed using the six-minute walk test, according to the American Thoracic Society (ATS) guidelines (2002), with a modification to suit clinical constraints. Due to limited indoor space and infection control concerns for immunosuppressed patients, a shortened 10 m walking course was used instead of the standard 30 m track [22]. Participants were instructed to walk at a self-paced speed for six minutes. Vital signs and Borg scale ratings for dyspnea and fatigue were recorded before and after each test. The tests were performed twice: preoperatively and at the end of week 4.

### 2.4. Spirometry

Spirometry was measured according to ATS/ERS guidelines [23], using a spirometer (Easy on-PC, ndd Medizintechnik AG, Zurich, Switzerland) in a seated position with a nose clip. The best of the three acceptable efforts was recorded. Variables included forced vital capacity (FVC), forced expiratory volume in one second/FVC ratio (FEV_1_/FVC), peak expiratory flow (PEF), forced expiratory flow at 25–75% of the pulmonary volume (FEV_25–75_), and percent predicted values. Testing was performed at baseline and week 4.

### 2.5. Respiratory Muscle Strength

Maximal inspiratory pressure (PImax) and maximal expiratory pressure (PEmax) were assessed using a portable manometer (MicroRPM, Micro Direct Inc., Lewiston, ME, USA) according to ATS/ERS recommendations [24]. Three reproducible maneuvers were performed for each test, and the highest values were recorded. Measurements were repeated seven times throughout the study period, once in week 1 and twice per week in weeks 2–4.

### 2.6. Lower Extremity Muscle Strength

Isometric strength of the quadriceps and tibialis anterior was assessed using a hand-held dynamometer (Lafayette Instrument Company, Lafayette, IN, USA) while patients were in the supine position. Three maximal voluntary contractions were performed per muscle group with a 5 s rest interval. The highest value (kg) was used in the analysis. Testing was performed seven times during the study period: once during pre-operative period and one or twice during weeks 1–4. For safety, the frequency of measurements depended on the participant’s condition.

### 2.7. Health-Related Quality of Life (HRQoL)

Quality of life was assessed using the Thai version of the Chronic Liver Disease Questionnaire (CLDQ), validated by Sobhonslidsuk et al. [25]. Only the “fatigue” and “activity” domains were analyzed. The questionnaire was administered twice weekly from week 1 to week 4. Each item on the CLDQ is scored using a 7-point Likert scale, where 1 indicates the worst frequency or severity (e.g., occurring “all of the time”) and 7 indicates no problems (e.g., “none of the time”). A higher score in either domain reflects better function or fewer symptoms.

### 2.8. Oxidative Stress Evaluation

Blood (3 mL) was drawn from the antecubital vein into EDTA tubes. Plasma from centrifuged blood (3000 rpm, 10 min) was analyzed for TAC and MDA within 3 h after collection. TAC was assessed by ABTS decolorization (734 nm) [26]. MDA was quantified using the TBARs method (532 nm) [27]. Data are expressed in mmol Trolx/L and µmol/L, respectively. Each measurement was conducted in triplicate and averaged.

### 2.9. Training Program

All the patients underwent rehabilitation under the supervision of a licensed physical therapist. During postoperative weeks 1 and 2, the patients followed a conventional treatment focused on respiratory hygiene, breathing exercise, and bed mobility. In postoperative weeks 3–4, patients transitioned to a structured exercise consisting of inspiratory muscle training (IMT), which was conducted twice daily using the POWERbreathe KH2 device (POWERbreathe International Ltd., Southam, UK). Training began at 40% PImax and progressed to 60%, targeting a Borg RPE of 3–5. Each session included 30 breaths (3 sets of 10) with a 1 min rest between sets, following Thammata et al. (2021) [28]. The early mobilization program was performed once daily using a progressive four-level protocol [28]. Exercise modalities included bedside stepping, hallway walking, and progressive ambulation, based on patient tolerance and monitored clinical signs. All training sessions were performed five days per week. The program is illustrated in Figure 1.

### 2.10. Statistical Methods

Descriptive statistics (mean ± SD) were used to summarize the participant characteristics and outcomes across the two rehabilitation phases: weeks 1–2 (conventional treatment) and weeks 3–4 (structured exercise). Changes over time were assessed using line graphs and individual-level analysis using the Brand–Middle Method and Bloom Table [29].

## 3. Results

### 3.1. Study Population

Patients who underwent LDLT were included because the procedure allows for planned rehabilitation and carries fewer acute risks than deceased donor transplantation. Due to limited donor availability in Thailand and the relatively low national LDLT case volume [30], only four patients were enrolled between 1 February 2024, and 31 January 2025. All the patients completed the full rehabilitation program without dropout. This study included two males and two females. The patients’ characteristics are presented in Table 1. Most patients were diagnosed with HCC. All patients have a Model for End-Stage Liver Disease (MELD) score of 13–16, which represents moderate risk. Mean body weight and BMI were both reduced (−11%) at the end of the experiment (week 4).

### 3.2. Cardiopulmonary Function

Cardiopulmonary function, including CPET, 6MWD, spirometry, and respiratory muscle strength before and after LT, as well as the following two distinct rehabilitation phases, are presented in Table 2 and Figure 2. Across all four cases, CPET showed a decrease in peak oxygen consumption (VO_2_ peak) in week 4. VO_2_ at the anaerobic threshold, VO_2_ peak, and VO_2_ peak% predicted decline in all patients. The 6MWD increased in Cases 1 (+31.9 m) and 4 (+253.0 m). Although spirometry showed a consistent postoperative FVC decline, FVC% predicted was <80% in Cases 1 and 3. FEV_1_/FVC ratios and % predicted improved in most cases. At week 4, all cases had improved expiratory flow parameters (PEF and FEF_25–75_), particularly Case 4. The PImax and PEmax decreased postoperatively in all cases. Structured exercise during weeks 3–4 led to clinically significant improvements in PImax in females (Cases 3 and 4) and PEmax in younger patients (Case 1 and 4).

### 3.3. Lower Extremity Muscle Strength

The results for lower-extremity muscle strength are shown in Figure 3. Quadriceps strength declined postoperatively in three patients. Whereas two patients had decreased TA strength. During weeks 3–4, structured exercise improved quadriceps and TA strength in female patients (Cases 3 and 4) and younger patients (Case 1 and 4), respectively.

### 3.4. Health-Related Quality of Life

Quality of life was assessed using the fatigue and activity domains of the CLDQ. The trends are illustrated in Figure 4. All patients expressed more fatigue and less activity immediately after the surgery. During structured exercise, patients expressed no change in fatigue index compared with conventional periods. However, two cases had a trend of improvement. None of the participants returned to their preoperative levels. Two patients improved the activity domain during the structured exercise period and regained their preoperative scores.

### 3.5. Oxidative Status

Antioxidant levels dropped after surgery. Structured exercise did not return serum TAC levels back to preoperative levels. After surgery, most patients had no change in serum MDA levels. Two patients presented a trend of decreased MDA during structured exercise. Serum TAC and MDA levels are shown in Figure 5.

## 4. Discussion

This case series was performed in four LDLT patients. The results showed post-transplant deconditioning across multiple domains, including respiratory muscle strength, cardiopulmonary fitness, self-evaluated fatigue and activity levels, and oxidative status. Following a two-week structured exercise program, favorable recovery trends were observed in respiratory muscle strength, quadriceps and tibialis anterior (TA) muscle strength, and self-reported activity levels in selected cases. However, a consistent reduction in total antioxidant capacity (TAC) was noted across all participants post-intervention.

Significant reductions in PImax and PEmax were observed in all patients after LT. These findings align with those reported by Siafakas et al. (2000) [31], who documented declines in PImax and PEmax 48 h after open cholecystectomy, another form of major upper abdominal surgery. Such acute impairments in respiratory muscle performance may stem from diaphragmatic and intercostal dysfunction due to postoperative inflammation, pain, anesthetic-related neuromuscular effects, and restricted thoracoabdominal movement. During the structural exercise program, PImax improved in women and PEmax improved in younger patients. Although PImax in male patients did not surpass trend-line expectations, these patients reached the minimal clinically important difference (MCID) of 17.2 cmH_2_O [32]. At the end of experiment, no patient had returned to their baseline PImax and PEmax levels. This observation is consistent with the findings of Deshmukh et al. (2017) [33], who implemented a two-week IMT following upper abdominal surgery. Although participants in their study exhibited significant improvements in PImax, they did not fully regain preoperative inspiratory muscle strength within the short follow-up period. This may highlight the challenges of early postoperative respiratory recovery. In our study, even two weeks of IMT and EM might not be sufficient for complete restoration. Sex- and age-related differences may influence outcomes. Females have a greater proportion of Type I diaphragm fibers (60–65% vs. 55–60% in males), conferring higher fatigue resistance despite lower absolute inspiratory strength [34]. Aging further reduces respiratory muscle strength, with PImax declining 0.8–2.7 cmH_2_O per year between the ages of 65 and 85 and diaphragm force being ~25% lower than that in young adults [35]. These factors may affect responses to inspiratory muscle training after liver transplantation

Persistent impairments in VO_2_ peak following LT have been reported [36]. Upper abdominal surgery may reduce exercise capacity due to respiratory muscle weakness, lower limb deconditioning, and inactivity. The reduction in VO_2_ peak observed in all patients may be attributable to diminished respiratory muscle strength (PImax and PEmax) and lung volume (FVC). Postoperative factors including physical inactivity, early postoperative fatigue, and the myopathic effects of calcineurin-inhibitor-based immunosuppressive regimens can further impair oxygen extraction and utilization at the muscle level. Additionally, postoperative complications and hepatopulmonary syndrome (HPS) can contribute to respiratory muscle weakness [31] and restrictive lung patterns [37]. Interestingly, cardiopulmonary exercise testing outcomes showed inconsistent patterns. Although VO_2_ peak declined uniformly, the 6MWD notably improved in two younger patients, surpassing the MCID range of 14 to 30.5 m reported for adults with various chronic conditions [38]. The increasing 6MWD in younger patients may be related to the reduced reduction of VO_2_ peak and improved TA muscle strength. However, the improvement may not be enough to recover cardiopulmonary fitness as measured by CPET. Achieving full recovery would require additional gains in respiratory muscle strength, lung function, and muscle strength. This suggests the need for extended intervention durations that include comprehensive respiratory and muscular rehabilitation. Paul et al. [39] highlighted that postoperative aerobic exercise performed over 4 to 12 weeks significantly improved functional capacity after intra-abdominal surgery, while Hegazy et al. [40] reported that high-intensity, long-duration IMT over eight weeks was necessary to restore and enhance pulmonary function tests (FVC, FEV_1_) beyond baseline levels after mitral valve replacement surgery.

It is not surprising that the self-evaluation fatigue and activity domains of the CLDQ worsened after LT. Fatigue levels remained largely unchanged, whereas activity scores improved marginally in only two cases. This lack of substantial improvement might result from postoperative abdominal discomfort, the presence of abdominal drainage tubes, and activity limitations imposed by immunosuppressive medication protocols restricting patient exposure.

Lower TAC levels and elevated MDA levels before and during transplantation are associated with increased risks of mortality and graft dysfunction. Previous studies suggest that decreased TAC and increased MDA [7] levels are due to ischemia and the reperfusion of liver tissue [6]. In this study, all patients displayed a trend of decreasing TAC compared to conventional treatment, although MDA levels decreased in two patients. This response may be attributed to tacrolimus, which is a commonly used immunosuppressant. Although the findings in patients after transplantation are not clear, tacrolimus may decrease parameters of oxidative stress, such as MDA, myeloperoxidase, and neutrophilic infiltration in vitro and animal studies [41]. Similar findings of lower TAC levels were reported in kidney transplant patients receiving tacrolimus compared to healthy controls [42].

This study has several limitations. Due to the small sample size, a design was employed to compare the treatment and control periods within each subject. Therefore, no concurrent control group received conventional treatment during the same period as the structured rehabilitation program (weeks 3–4). This lack of a concurrent comparator limits the ability to attribute the observed improvements specifically to the intervention. These findings reflect the outcomes observed in a limited sample of four individual cases and should be interpreted with caution.

## 5. Conclusions

This study suggests that a two-week structural rehabilitation program composing IMT and progressive EM might not be sufficient to fully restore cardiopulmonary capacity after LT. However, the fact that several indicators were improved or maintained in selective cases, including 6MWD, respiratory muscle strength, and lower-limb muscle strength, demonstrates clinically meaningful recovery trajectories. Certain parameters, such as VO_2_ peak and FVC percentage predicted, may require longer rehabilitation durations to achieve significant physiological recovery. Future randomized controlled trials with larger cohorts are recommended to validate the efficacy of early structed rehabilitation strategies after LT.

## Figures and Tables

**Figure 1 jfmk-10-00313-f001:**
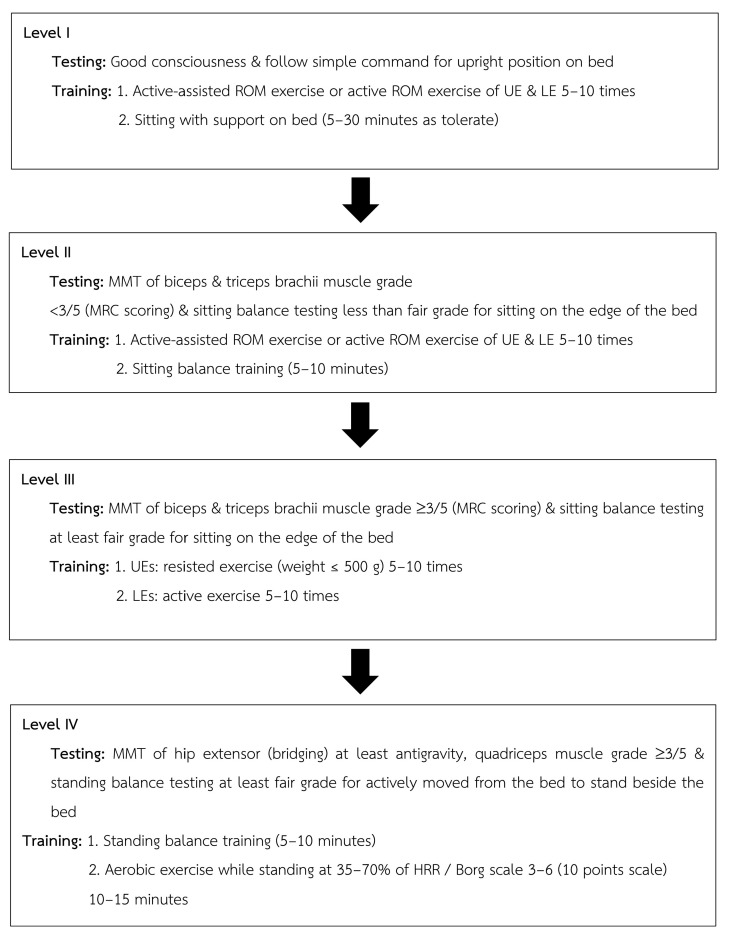
Progressive early mobilization used in the structural exercise program.

**Figure 2 jfmk-10-00313-f002:**
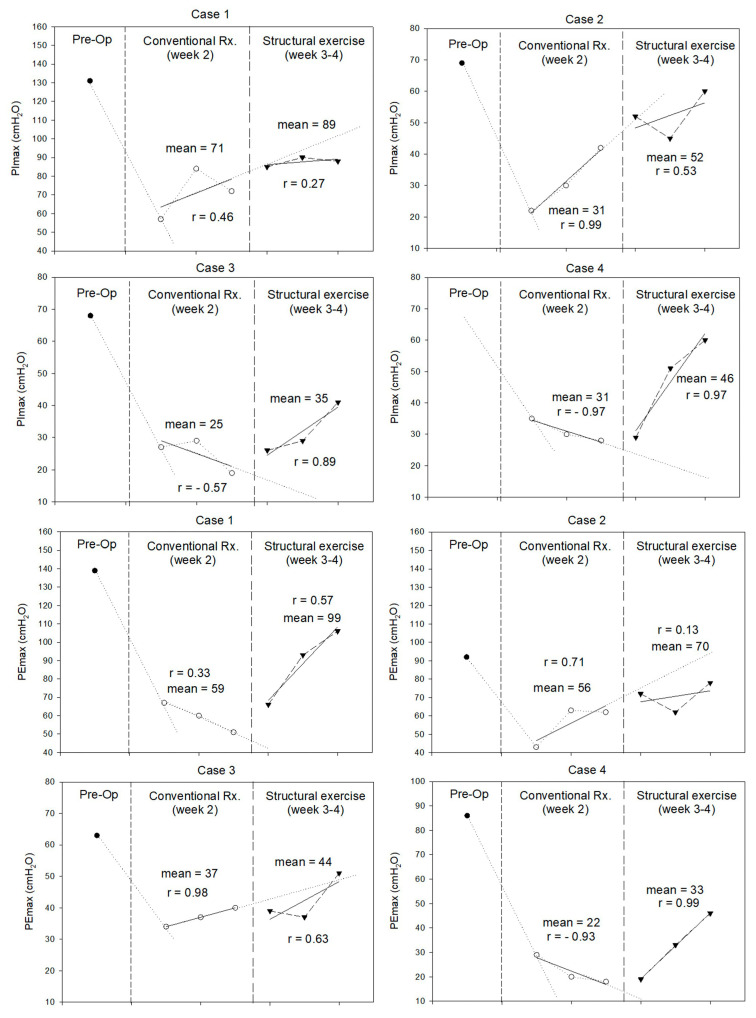
Maximal inspiratory pressure (PImax) and maximal expiratory pressure (PEmax) (in cmH_2_O) measured during pre-operation, conventional treatment, and structural exercise of each case.

**Figure 3 jfmk-10-00313-f003:**
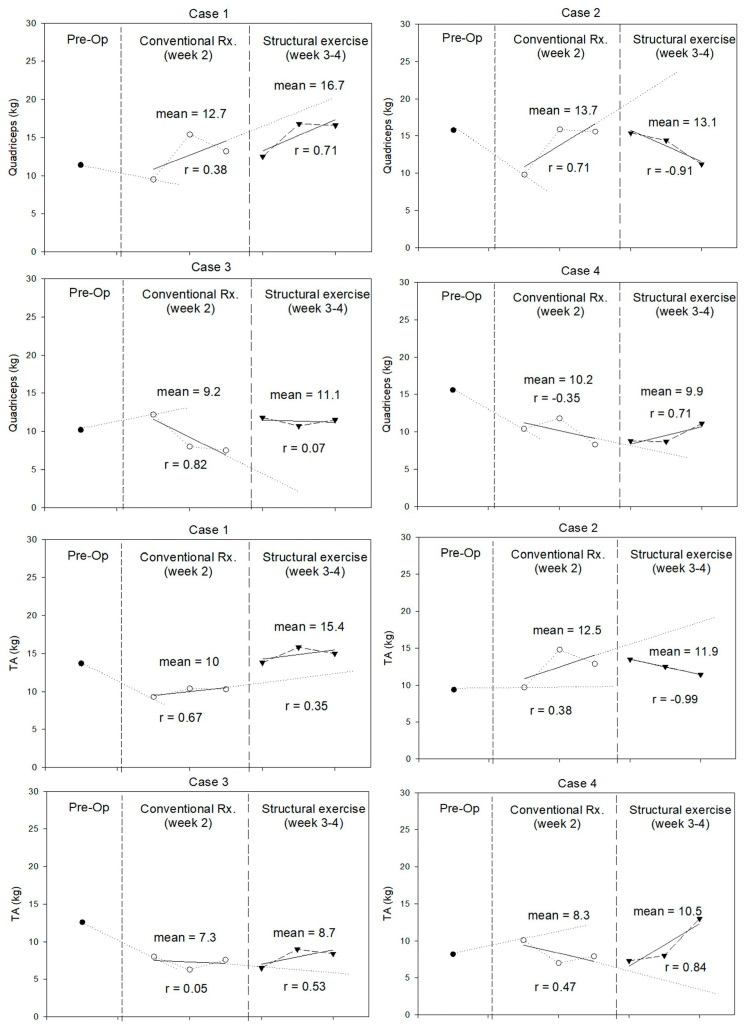
Quadriceps and tibialis anterior (TA) strength (in kilograms) measured during pre-operation (Pre-Op), conventional treatment, and structured exercise.

**Figure 4 jfmk-10-00313-f004:**
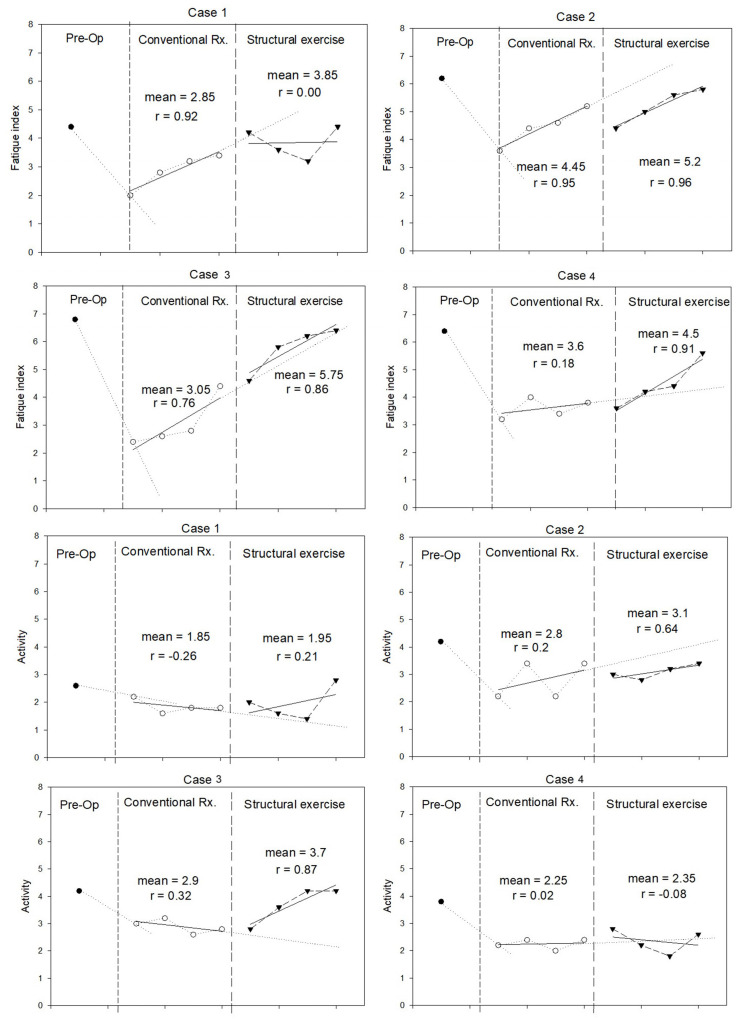
Fatigue and activity domains of the Chronic Liver Disease Questionnaire (CLDQ) (in score) measured during pre-operation, conventional treatment, and structural exercise for each case.

**Figure 5 jfmk-10-00313-f005:**
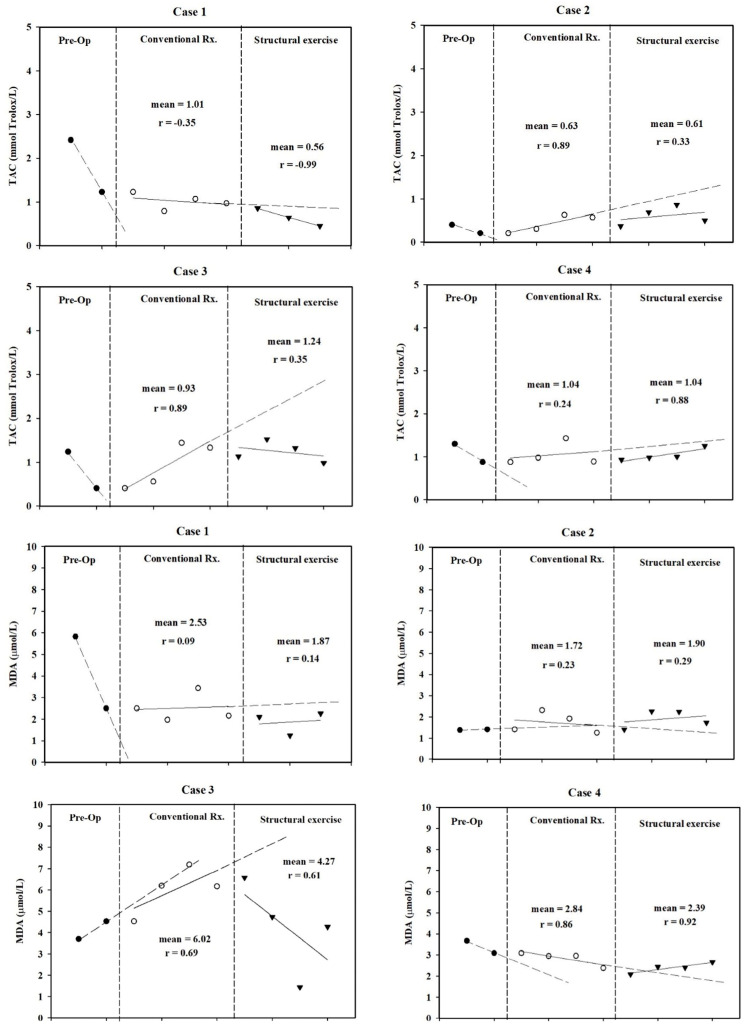
Total antioxidant capacity (TAC) and malondialdehyde (MDA) levels were measured at three time points pre-operation, during conventional treatment, and during structured exercise for each case.

**Table 1 jfmk-10-00313-t001:** Demographic and clinical characteristics of the liver transplant recipients.

	Case No.1	Case No.2	Case No.3	Case No.4	Mean ± SD
Gender, M/F	M	M	F	F	
Age, y	52	60	59	21	48.00 ± 18.35
Diagnosis	HCC, HBV	HCC, Alcoholic cirrhosis	HCC, HCV	Biliary atresia s/p Kasai operation	-
Preoperative MLED (Score)	14	13	16	13	14.00 ± 1.22
Underlying disease	HTN, DLP	-	-	VSD	-
Height, cm	158	166.5	155	155	158.62 ± 5.44
Weight, kg					
week-0	76	63	53.3	52	61.08 ± 11.09
week-4	67	53	53	44	54.25 ± 9.50
BMI (km/m^2^)					
week-0	30.44	22.73	22.19	21.64	24.25 ± 4.15
week-4	26.84	19.12	22.06	18.3	21.58 ± 3.86

Legend: M: male; F: female; HCC: hepatocellular carcinoma; HBV: hepatitis B virus; HCV: hepatitis C virus; s/p: status post; Preoperative MLED: Model for End-Stage Liver Disease score before surgery; BMI: body mass index.

**Table 2 jfmk-10-00313-t002:** Cardiopulmonary exercise testing, 6 min walk distance, and spirometry of patients before and after intervention.

	Case No.1	Case No.2	Case No.3	Case No.4	Mean ± SD
**CPET**					
**VO_2_ at AT, mL/kg/min**					
week-0	14.9	18.7	7.1	10.7	12.85 ± 5.04
week-4	11.3	15.5	6.6	12.7	11.52 ± 3.72
**VO_2_ peak, mL/kg/min**					
week-0	20.6	24.6	16	22.6	20.95 ± 3.68
week-4	14.8	19.8	9	16.5	15.02 ± 4.52
**VO_2_ peak %Pred, %**					
week-0	75	60	32	65	58.00 ± 18.42
week-4	47	42	18	44	37.75 ± 13.33
**Peak workload, Watts**					
week-0	122	142	25	116	101.25 ± 52.03
week-4	91	83	25	82	70.25 ± 30.43
**6MWD, m**					
week-0	344.1	490	365	195	348.52 ± 120.94
week-4	376	414	207.2	448	361.30 ± 106.86
**Spirometry**					
**FVC (L)**					
week-0	2.53	4.8	1.86	3.41	3.15 ± 1.27
week-4	2.51	4.59	1.45	2.47	2.76 ± 1.32
**FVC %Pred (%)**					
week-0	78	144	80	119	105.25 ± 31.99
week-4	78	138	62	86	91.00 ± 32.88
**FEV_1_/FVC (%)**					
week-0	81.1	67.6	75.7	91.6	79.00 ± 10.07
week-4	80.5	70.3	82.8	97	82.65 ± 11.00
**FEV_1_/FVC %Pred (%)**					
week-0	96	82	90	101	92.25 ± 8.18
week-4	95	86	98	107	96.50 ± 8.66
**PEF (L)**					
week-0	7.59	9.35	4.07	5.76	6.69 ± 2.28
week-4	8.44	9.85	4.24	5.88	7.10 ± 2.52
**PEF %Pred (%)**					
week-0	91	110	71	94	91.50 ± 16.01
week-4	102	116	77	96	97.75 ± 16.17
**FEV_25–75_ (L)**					
week-0	1.83	1.74	1.19	3.5	2.06 ± 1.00
week-4	1.97	1.95	1.3	3.9	2.28 ± 1.12
**FEV_25–75_ %Pred (%)**					
week-0	54	55	54	104	66.75 ± 24.84
week-4	58	62	59	115	73.50 ± 27.11

Legend: CPET: cardiopulmonary exercise testing; VO_2_ at AT: oxygen uptake at anaerobic threshold; VO_2_ peak: peak oxygen uptake; %pred: percent of predicted value; 6MWD: six-minute walk distance; FVC: forced vital capacity; FEV_1_/FVC: forced expiratory volume in one second/FVC ratio; PEF: peak expiratory flow; FEV_25–75_: forced expiratory flow between 25% and 75% of FVC; %: percentage.

## Data Availability

The data presented in this study are available on request from the corresponding author. The data are not publicly available due to restrictions related to patient privacy and ethical considerations in accordance with the Declaration of Helsinki and the approval of the Research Ethics Committee, Faculty of Medicine, Chiang Mai University (Approval Code: NONE-2566-0422), in order to ensure that subsequent use of the data is appropriate and does not infringe upon the rights of the study participants.

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
