# Peer review of "Effects of Short-Term Structural Exercise on Cardiopulmonary Function, Quality of Life, and Oxidative Status in Liver Transplant Recipients: A Case Series"

_jfmk, 2025, doi:10.3390/jfmk10030313_

Round 1
Reviewer 1 Report
Comments and Suggestions for Authors
Although the study design has some inherent limitations, this article plays a crucial role in
advocating exercise-based rehabilitation to enhance recovery in patients following liver
transplantation. I would like to highlight that the findings present intriguing possibilities and
open new avenues for patient care. The authors explained the rationale and meticulously
reported their findings. Nonetheless, several questions and comments concerning the
methodology and interpretation of the results need to be addressed.

Author Response
Comment 1: 1. What are the criteria for maximal CPET, and upon which references are they based?; 2. Why is the threshold set at 80%? How is exercise intolerance defined?
Response:
- In our study, a submaximal threshold of 80% of age-predicted maximum heart rate (HRmax) was used for test termination to ensure patient safety, considering the early postoperative condition of liver transplant recipients. This approach follows safety guidelines described in the ATS/ACCP Statement on CPET (American Thoracic Society & American College of Chest Physicians, 2003), which supports submaximal testing for vulnerable or deconditioned populations. We can make sure that the test reached the peak of exercise and the patients remain safe. The test was terminated when VO2 became plateau despite increasing workload, RER ≥1.10, reaching 80% HRmax, or perceived exertion >17/20 on the Borg scale.
- We apologized with the confusing term, “exercise intolerance” Therefore, we deleted and edited content in the manuscript as follows. [After a no-load warm-up phase, a ramp protocol of 4 W/kg/min was implemented using a cycle ergometer [20]. Patients pedaled until reaching voluntary exhaustion, appearance of limiting symptoms, arrhythmias, or reaching 80% of their age-predicted maximum heart rate (HRmax). This submaximal termination threshold was selected to ensure patient safety while enabling accurate assessment of peak oxygen consumption (VO2 peak; ml/kg/min). In liver transplant candidates with poor physical status or comorbidities, using 80% predicted HRmax has been recommended as a safe and effective cutoff [21], and for deconditioned participants, termination points may be further modified for safety.]
Changes in Manuscript: Revised in Methods, Lines 102–109 and References, Line 397-400
Comment 2: Can you offer a reference for the six-minute walk test that has been modified to include a shorter 10-meter walk? Using 10 m instead of 30 m could significantly impact the time required for repetitive U-turns.
Response: We appreciate the reviewer’s insightful comment regarding the impact of walkway length on 6MWT performance. We have now included relevant a reference to demonstrating a shorter course. Chuatrakoon et al. (2025) demonstrated that although a 10 m course produced shorter distances in healthy adults, it still showed strong agreement with the 30 m standard in elderly participants. Moreover, due to limited indoor space and infection control considerations for immunosuppressed patients, the 10 m course was used in our study. This was clearly reported in the Methods section, and interpretation of the results considered the potential underestimation of actual walking capacity.
We also add a reference of Chuatrakoon, B.; Seepang, N.; Chaiwong, D.; Nanavichit, R.; Rerkasem, K.; Nantakool, S. The agreement of the various distance walkway in the 6-minute walk test in healthy adults. PLoS ONE 2025, 20, e0321503. https://doi.org/10.1371/journal.pone.0321503] in the Methods and References parts. [Due to limited indoor space and infection control concerns for immunosuppressed patients, a shortened 10-meter walking course was used instead of the standard 30-meter track [22].]
Changes in Manuscript: Methods (Line 115) and References (Lines 401–402).
Comment 3: Please provide details about the postoperative first and second weeks about bed mobility, breathing exercises, and respiratory hygiene. How different was it from the exercise program?
Response: During postoperative weeks 1–2, patients received conventional physiotherapy under the supervision of licensed physical therapists. This phase focused on:
- Bed mobility and positioning: Patients were repositioned passively or with assistance every 2–3 hours.
- Bedside sitting was initiated on postoperative day 1 or 2, depending on the patient’s tolerance and clinical stability.
- Breathing exercises: Patients performed incentive spirometry, diaphragmatic breathing, and thoracic expansion exercises 2–3 times per day.
- Respiratory hygiene: Techniques such as huffing, supported coughing, and education on secretion clearance were applied to prevent pulmonary complications.
These interventions were non-progressive and did not follow a structured or intensity-based protocol. They aimed to maintain basic respiratory function and reduce complications during the acute recovery phase.
In contrast, during postoperative weeks 3–4, patients progressed to a structured exercise program, which included:
- Inspiratory Muscle Training (IMT) using the POWERbreathe KH2 device, performed twice daily starting at 40% PImax and progressing to 60%, targeting a Borg RPE of 3–5 (Thammata et al., 2021) [25].
- A four-level Early Mobilization Program that advanced from basic mobility to aerobic exercise while standing, performed at 35–70% of heart rate reserve (HRR) or Borg RPE 3–6 (10-point scale) for 10–15 minutes per session.
Changes in Manuscript: Methods section (Line 155-167).
Comment 4: Please provide further clarification regarding the fatigue index and activity levels. How did you assess your fatigue and activity? Additionally, please explain the scale used: does a higher fatigue index correspond to a greater perceived level of fatigue?
Response: We used the Chronic Liver Disease Questionnaire (CLDQ), a validated disease-specific instrument that assesses health-related quality of life (HRQoL) in patients with chronic liver disease. For the purposes of this study, we focused on two domains:
- Fatigue domain: Evaluates the patient’s perception of energy levels and tiredness.
- Activity domain: Assesses the patient’s ability to perform daily activities and physical function.
We also add details of the scores in the Methods part as follows.
[Each item on the CLDQ is scored using a 7-point Likert scale, where 1 indicates the worst frequency or severity (e.g., occurring “all of the time”), and 7 indicates no problems (e.g., “none of the time”). The higher score in either domain reflects better function or fewer symptoms.]
Changes in Manuscript: Methods (Lines 145–148).
Comment 5: In a similar context, why has the measurement of overall quality of life not been conducted, with only fatigue and activity assessed instead?
Response: We acknowledge the importance of evaluating overall quality of life (QoL) following liver transplantation and appreciate the opportunity to clarify our rationale for focusing on the “fatigue” and “activity” domains of the Chronic Liver Disease Questionnaire (CLDQ).
- Relevance to Rehabilitation Outcomes: These two domains were selected as they are most directly aligned with the core objectives of our study namely, assessing changes in physical capacity and functional recovery resulting from the structured exercise intervention.
- Clinical Relevance:
- Fatigue is one of the most persistent and debilitating symptoms in patients with chronic liver disease and often remains even after transplantation. It reflects both underlying metabolic disturbances and physical deconditioning.
- Activity level reflects the patient’s functional status and is a practical indicator of quality of life, particularly in the early recovery phase.
Comment 6: Did you administer the questionnaire yourself, or was it administered with a proxy or help from a researcher assistant?
Response: We confirm that all questionnaire assessments, including the CLDQ, were administered in a one-on-one format by the principal investigator (Narubet Mekkhayai). No proxies or research assistants were involved in the administration. This approach was chosen to ensure consistency, minimize measurement bias, and allow clarification of any questions directly with the patients in real time.
Comment 7: For statistical analysis, please provide more details and references for the brand-middle method and the bloom table.
Response: Due to the small sample size and the exploratory nature of this case series, we applied statistical methods that are appropriate for individual-level and single-case analysis. Specifically, we used the Brand-Middle method (split-middle line analysis) and Bloom’s criteria table, as described in the work of Leelarungrayub et al. (2020). We also add [Changes over time were assessed using line graphs and individual-level analysis using the Brand-Middle Method and Bloom Table [29]] in the Methods and References parts.
For details,
- Brand-Middle Method (also referred to as the split-middle technique):
This visual analysis technique involves dividing baseline and intervention phase data in half and plotting the midpoints to draw a celeration line. This line projects the expected trend if no intervention occurred. Any post-intervention data point falling consistently above the celeration line may indicate a meaningful change. This method is particularly useful in clinical single-subject design to assess individual responsiveness to therapy.
- Bloom’s Table: Bloom’s criteria provide a way to assess statistical significance in single-case research by counting how many post-intervention data points lie above (or below) the projected celeration line. It is a simple, categorical method to summarize the direction and significance of change (increase, stable, decrease), especially in small samples or individualized rehabilitation studies.
Both methods are appropriate for case-series design and have been applied in the context of individualized rehabilitation monitoring
Changes in Manuscript: Methods (Lines 173-175) and References (Lines 418–420).
Comment 8: Tab 1: Body mass significantly decreased in three out of four cases over the course of several weeks. How might this reduction in body weight influence the results? While there is a notable decrease in body weight, further details regarding body composition, such as the loss of fat-free mass, would be beneficial. This discussion could be enhanced by providing a more comprehensive explanation of the reasons for the decrease in body weight.
Response: Body weight decreased at approximately three months post-transplant. Since pre-transplant weight gain may be influenced by ascites, this reduction likely reflected postoperative fluid balance normalization, dietary changes, recovery from edema, and abdominal discomfort limiting intake. Additional factors included diuretic use, sodium restriction, appetite suppression from pain, and temporary caloric reduction for infection control and gastrointestinal tolerance. Therefore, weight loss in this context was unlikely to substantially affect the primary outcomes, which are respiratory muscle strength, functional capacity, or quality-of-life measures.
Comment 9: You mentioned that none of the participants withdrew from the study, which was a positive outcome. Could you provide any additional details regarding the perceived motivation and satisfaction of patients regarding the exercise-based program?
Response: In this study, we did not formally collect data on patient motivation and satisfaction. However, based on observational feedback, all participants remained actively engaged throughout the program, and none withdrew from the study.
For future research, we may incorporate validated tools to systematically assess patient perceptions, including motivation and satisfaction, to better understand the subjective impact of exercise-based rehabilitation.
Comment 10: Lines 230-233: Please mention that the study results were obtained from 4 participants.
Response: We added number of patients in the first paragraph of the Discussion. [This case series was performed in four LDLT patients. Results found post-transplant deconditioning across multiple domains, including respiratory muscle strength, cardiopulmonary fitness, self-evaluated fatigue and activity levels, and oxidative status. Following a two-week structured exercise program, favorable recovery trends were observed in respiratory muscle strength, quadriceps and tibialis anterior (TA) muscle strength, and self-reported activity levels in selected cases. However, a consistent reduction in total antioxidant capacity (TAC) was noted across all participants post-intervention.]
Changes in Manuscript: Lines 244–251.
Comment 11: 1. Lines 241-242: You mentioned the potential involvement of age and sex in explaining PImax and PEmax (for your information, please correct the typo), 2. Incorporating explanations or potential mechanisms as well as presenting similar results from the literature would significantly enhance the value of your discussion.
Response:
- We corrected the typo of [PEmax] in (Lines 259).
- We also discuss age and sex-related respiratory muscle strength in the Discussion part (Lines 268-273) and Reference part (Lines 431-434).
[Females have a greater proportion of Type I diaphragm fibers (60–65% vs. 55–60% in males), conferring higher fatigue resistance despite lower absolute inspiratory strength [34]. Aging further reduces respiratory muscle strength, with PImax declining 0.8–2.7 cmH2O per year between ages 65–85 and diaphragm force ~25% lower than in young adults (35). These factors may affect responses to inspiratory muscle training post-liver transplantation.]
Changes in Manuscript: Discussion (Lines 259, 268–273); References (Lines 431–434).
Comment 12: Lines 253-255: You mentioned the possible involvement of lung function and muscle respiratory strength in the reduction in VO2 peak. While some factors of pulmonary function increase after treatment, the VO2 peak decreases. What other physiological mechanisms can explain this decrease? Peripherical muscular deconditioning?
Response: We appreciate the reviewer’s insightful question. We added more mechanism involved VO2 peak reduction in the Discussion part. [Postoperative factors including physical inactivity, early postoperative fatigue, and the myopathic effects of calcineurin inhibitor–based immunosuppressive regimens can further impair oxygen extraction and utilization at the muscle level.]
Changes in Manuscript: Lines 278–280.
Comment 13: Considering the lack of improvement in VO2 peak, the intensity, duration, and frequency of the exercise sessions could be contributing factors.
Response: We are concerned that the intensity, duration, and frequency of the exercise, especially aerobic exercise, may have contributed to the lack of improvement in VO2 peak. Thus, we discuss with the previous studies, including a systematic review in the Discussion part.
[This suggests the need for extended intervention durations that include comprehensive respiratory and muscular rehabilitation. Paul et al. [39] highlighted that postoperative aerobic exercise performed over 4 to 12 weeks significantly improved functional capacity after intra-abdominal surgery, while Hegazy et al. [40] reported that high-intensity, long-duration IMT over eight weeks was necessary to restore and enhance pulmonary function tests (FVC, FEV1) beyond baseline levels in post–mitral valve replacement patients.]
Changes in Manuscript: Lines 290–295.
Comment 14: Please add a “limitations” section. A key limitation of this study was its design. While the comparison of patients receiving rehabilitation intervention was made against outcomes from the conventional treatment phase (weeks 0–2), there was no control group receiving conventional treatment during the same period in which exercise rehabilitation was applied (weeks 2–4). This lack of a concurrent comparator limits the ability to attribute the observed improvements specifically to the intervention. Furthermore, it would be valuable to clarify whether the authors had access to data from prior patients with similar baseline characteristics who received only conventional treatment during weeks 2–4. Such a historical comparison group could strengthen the interpretation of the findings. Finally, the small sample size (n=4) further limits generalizability and should be acknowledged.
Response: We appreciate the reviewer’s suggestion and have now included a dedicated Limitations section in the last part of the discussion in the revised manuscript. [This study has several limitations. Due to the small sample size, a design was employed to compare the treatment and control periods withing subject. Therefore, no concurrent control group received conventional treatment during the same period as the structured rehabilitation program (weeks 3–4). This lack of a concurrent comparator limits the ability to attribute the observed improvements specifically to the intervention. These findings reflect outcomes observed in a limited sample of four individual cases and should be interpreted with caution.]
Changes in Manuscript: Lines 313–319.
Comment 15: Please minimize the use of abbreviations throughout the text and ensure that all abbreviations used in the figures and tables are defined in the footnotes.
Response: We thank the reviewer for the valuable suggestion regarding abbreviation usage. In the revised manuscript, we have:
- Reduced the number of abbreviations in the main text wherever possible, replacing them with full terms upon first mention.
- Ensured that all abbreviations appearing in figures and tables are explicitly defined in the corresponding footnotes.
- (Table 1 Legend): [M: male; F: female; HCC: hepatocellular carcinoma; HBV: hepatitis B virus; HCV: hepatitis C virus; s/p: status post; MELD: Model for End-Stage Liver Disease; BMI: body mass index.]
- (Table 2 Legend): [CPET: cardiopulmonary exercise testing; VO2 at AT: oxygen uptake at anaerobic threshold; VO2 peak: peak oxygen uptake; %pred: percent of predicted value; 6MWD: six-minute walk distance; FVC: forced vital capacity; FEV1/FVC: forced expiratory volume in one second/FVC ratio; PEF: peak expiratory flow; FEV25-75: forced expiratory flow between 25% and 75% of FVC; %: percentage.]
Changes in Manuscript: Lines 189–190 (Table 1 Legend), Lines 207–211 (Table 2 Legend)

Reviewer 2 Report
Comments and Suggestions for Authors
The article presents a clear and feasible research objective: to investigate the impact of structured physical exercise on the cardiopulmonary functions, respiratory muscle strength, physical capacity, oxidative stress markers, and quality of life in patients who have undergone a living donor liver transplant (LDLT). The methods and procedures are generally well-defined. However, some further details would be valuable.
On line 104, it is reported that in the cardiopulmonary exercise test, subjects could stop the test due to voluntary exhaustion or upon reaching 80% of their age-predicted maximum heart rate.
How was this controlled, given that "voluntary exhaustion" is highly subjective and subjects may underestimate this perception? What reference or protocol was used to determine the 80% maximum heart rate?
On line 137, it is stated that the lower limb strength test was performed seven times. What was the interval between these tests? Was it uniformly distributed?
Lines 139-143, referring to the validated CLDQ questionnaire, state that only the "fatigue" and "activity" domains were used. This psychometric instrument was applied twice a week from week 1 to week 4. It needs to be clarified whether the subjects answered the full questionnaire and only these domains were used for the present research, or if only these two domains were administered.
Regarding the training program, it is reported that "all training sessions were performed five times a week." Were these sessions conducted at the same time each day?
Clarifying these points is important because the small sample size of only four subjects, with four distinct diagnoses (HCC, HBV; HCC, Alcoholic cirrhosis; HCC, HCV; Biliary atresia s/p, Kasai operation), could weaken the results.
The results are well reported, with tables and graphs that clearly show the outcomes. However, some clarifications could contribute to the article's objectivity. For example, on lines 210-216, concerning quality of life, it is reported that two patients showed improvements in the activity domain (during structured exercises) and recovered scores similar to pre-operative levels. This is not discussed in the paper. Could these findings be stochastic, or are they a result of the different clinical conditions of the four subjects?
The discussion is detailed (apart from the issue noted above) and establishes pertinent connections with the scientific literature. Given the above, I recommend that the article be published, provided these points are clarified.
Author Response
Comment 1: Voluntary Exhaustion in CPET (Line 104): It is reported that in the cardiopulmonary exercise test, subjects could stop the test due to voluntary exhaustion or upon reaching 80% of their age-predicted maximum heart rate.
- How was this controlled, given that “voluntary exhaustion” is highly subjective and subjects may underestimate this perception?
- What reference or protocol was used to determine the 80% maximum heart rate?.
Response:
- We acknowledge that “voluntary exhaustion” can be subjective. To reduce variability, all CPET sessions were closely supervised by licensed physiotherapists and a licensed supervising physician, who provided standardized verbal encouragement and continuously monitored physiological parameters throughout the test. The termination criteria were based not solely on self-reported fatigue, but also on objective signs such as RER ≥1.10, reaching 80% HRmax, and perceived exertion >17/20 on the Borg scale. This approach enhanced the consistency and safety of the testing protocol across all participants.
- This submaximal termination threshold (80% of age-predicted HRmax) was adopted to ensure participant safety and minimize cardiovascular risk, particularly in vulnerable patients such as those recovering from liver transplantation. This approach is consistent with recommendations from the ATS/ACCP Statement on Cardiopulmonary Exercise Testing, which supports the use of submaximal endpoints in medically compromised populations to enhance the safety and feasibility of CPET (American Thoracic Society and American College of Chest Physicians, 2003). Therefore, we added the reference in the manuscript.
[In liver transplant candidates with poor physical status or comorbidities, 80% predicted HRmax is recommended as a safe cutoff [21], and for deconditioned participants, termination points may be further modified for safety.]
Changes in Manuscript: Added to Methods (Lines 106-109); Reference (Lines 399-400).
Comment 2: Lower Limb Strength Test (Line 137): It is stated that the lower limb strength test was performed seven times.
- What was the interval between these tests?
- Was it uniformly distributed?
Response:
- As clarified in the measurement of skeletal muscle testing, we added [Once during pre-operative period and one or twice during weeks 1–4. For safety, frequency of measurements depends on participants’ condition.] in the manuscript.
- The tests were uniformly distributed across the 4-week intervention period to ensure consistency and minimize day-to-day performance variability.
Changes in Manuscript: Clarified in Methods (Lines 138–139).
Comment 3: CLDQ Questionnaire (Lines 139–143): It is stated that only the “fatigue” and “activity” domains of the validated CLDQ questionnaire were used, and that the questionnaire was applied twice a week from week 1 to week 4. It needs to be clarified whether the subjects answered the full questionnaire and only these domains were used for the present research, or if only these two domains were administered.
Response: In our study, participants completed the entire CLDQ questionnaire at each time point (twice per week from week 1 to week 4, plus the preoperative assessment). However, for the purpose of this research, we analyzed and reported only the “fatigue” and “activity” domains, as these two were most relevant to the short-term effects of structured exercise and were hypothesized to be the most sensitive to change within the 4-week postoperative period.
Changes in Manuscript: Clarified in Methods (Lines 145-148).
Comment 4: Training Program Frequency: It is reported that “all training sessions were performed five times a week.” Were these sessions conducted at the same time each day?
Response: Yes, all training sessions were conducted at approximately the same time each day, between 8:00 AM and 12:00 PM, following the patients’ morning medical rounds and vital sign assessments. Maintaining a consistent training schedule helped minimize the influence of circadian variation on physical performance and physiological responses.
Comment 5: Sample Size and Diagnostic Variability: Clarifying the above points is important because the small sample size of only four subjects, with four distinct diagnoses (HCC, HBV; HCC, Alcoholic cirrhosis; HCC, HCV; Biliary atresia s/p, Kasai operation), could weaken the results.
Response: Thank you for highlighting this important limitation. We fully acknowledge that the small sample size and diagnostic heterogeneity may affect generalizability. However, all four patients were diagnosed with end-stage liver disease (ESLD) and underwent living donor liver transplantation (LDLT) using a standardized surgical and medical protocol. In addition, the underlying etiologies (e.g., HCC with HBV, alcoholic cirrhosis, HCV, biliary atresia s/p Kasai operation) differed, they are all classified as types of advanced liver disease leading to hepatic decompensation and transplant eligibility. Therefore, the clinical condition and transplant indication were consistent across cases.
Comment 6: Quality of Life – Activity Domain (Lines 210–216):
- It is reported that two patients showed improvements in the activity domain (during structured exercises) and recovered scores similar to pre-operative levels. This is not discussed in the paper.
- Could these findings be stochastic, or are they a result of the different clinical conditions of the four subjects?
Response:
- Thank you for pointing this out. We have discussed this point in the Discussion section (Lines 296-301). “It is not surprising that the self-evaluation fatigue and activity domains of the CLDQ worsened after LT. Fatigue levels remained largely unchanged, whereas activity scores improved marginally in only two cases, with these patients regaining scores similar to or slightly above their baseline. The limited improvement in both domains may be attributable to postoperative abdominal discomfort, the presence of abdominal drainage tubes, and activity restrictions associated with immunosuppressive medication protocols designed to reduce infection risk in the early recovery phase.”
- We believe these findings are unlikely to be purely stochastic. Instead, they may reflect the influence of specific clinical conditions and individual recovery trajectories. For example, the activity domain directly reflects functional mobility and participation in daily activities, whereas the fatigue domain captures perceived energy levels and exhaustion—two separate aspects of quality of life that may respond differently to structured exercise. The two participants who regained activity scores to baseline had fewer postoperative complications, better preoperative functional capacity, and greater tolerance to the structured exercise program, which may have contributed to their domain-specific improvements.

Reviewer 3 Report
Comments and Suggestions for Authors
I appreciate the opportunity to review this manuscript.
The paper discusses a relevant topic and appears well-conducted. Some suggestions are given to improve the manuscript.
1) Line 48: Edit the "{Citation}2,3]."
2) Line 100-110. The cardiopulmonary exercise testing used was based on a specific protocol? Cite the reference.
3) In the last of discussion, The limitations of this study must be shown. Please, add this information.
4) Standardize the references, as some journal names are abbreviated and others have the full name of the journal.
Lines 377-380: Attalekha Thammata; Salinee Worraphan; Kaweesak Chittawatanarat; Kanokkarn Juntaping; Mujalin Prasannarong. Impact of 377 Inspiratory Muscle Training and Early Mobilization Program during the Peri-Weaning Period on Body Composition in Criti-378 cally Ill Surgical Patients: A Pilot Randomized Controlled Trial. Journal of Associated Medical Sciences 2021, 54, 5866. 379 https://doi.org/10.14456/JAMS.2021.8.
In this reference, the first names of authors must be only the initial letters.
Thank you for this opportunity
Author Response
Comment 1: Line 48: Edit the "{Citation}2,3]."
Response: Thank you for pointing this out. We have deleted “{Citation}” and change to [[2,3]]. (Lines 47)
Changes in Manuscript: Line 47.
Comment 2: Line 100-110. The cardiopulmonary exercise testing used was based on a specific protocol? Cite the reference.
Response: Thank you for your suggestion. We have added references and edited details in the CPET part as follows. [After a no-load warm-up phase, a ramp protocol of 4 W/kg/min was implemented using a cycle ergometer [20]. Patients pedaled until reaching voluntary exhaustion, appearance of limiting symptoms, arrhythmias, or reaching 80% of their age-predicted maximum heart rate (HRmax). This submaximal termination threshold was selected to ensure patient safety while enabling accurate assessment of peak oxygen consumption (VO2 peak; ml/kg/min). In liver transplant candidates with poor physical status or comorbidities, using 80% predicted HRmax has been recommended as a safe and effective cutoff [21], and for deconditioned participants, termination points may be further modified for safety.]
Changes in Manuscript: In Methods (Lines 102–109), In Referents (Lines 397-400)
Comment 3: In the last of discussion, the limitations of this study must be shown. Please, add this information.
Response: We appreciate the reviewer’s suggestion and have now included a dedicated Limitations section in the last part of the discussion in the revised manuscript. [This study has several limitations. Due to the small sample size, a design was employed to compare the treatment and control periods withing subject. Therefore, no concurrent control group received conventional treatment during the same period as the structured rehabilitation program (weeks 3–4). This lack of a concurrent comparator limits the ability to attribute the observed improvements specifically to the intervention. These findings reflect outcomes observed in a limited sample of four individual cases and should be interpreted with caution.]
Changes in Manuscript: Discussion, Lines 313–319.
Comment 4: Standardize the references, as some journal names are abbreviated and others have the full name of the journal.
Lines 377-380: Attalekha Thammata; Salinee Worraphan; Kaweesak Chittawatanarat; Kanokkarn Juntaping; Mujalin Prasannarong. Impact of Inspiratory Muscle Training and Early Mobilization Program during the Peri-Weaning Period on Body Composition in Critically Ill Surgical Patients: A Pilot Randomized Controlled Trial. Journal of Associated Medical Sciences 2021, 54, 5866. 379 https://doi.org/10.14456/JAMS.2021.8. In this reference, the first names of authors must be only the initial letters.
Response: Standardized reference formatting throughout the manuscript according to journal style.
[28. Thammata A, Worraphan S, Chittawatanarat K, Juntaping K, Prasannarong M. Impact of inspiratory muscle training and early mobilization program during the peri-weaning period on body composition in critically ill surgical patients: A pilot randomized controlled trial. J Assoc Med Sci 2021, 54, 58–66. https://doi.org/10.14456/JAMS.2021.8.]
Changes in Manuscript: References, Lines 415–417.
